# Structure, Functions, and Implications of Selected Lipocalins in Human Disease

**DOI:** 10.3390/ijms25084290

**Published:** 2024-04-12

**Authors:** Preethi Chandrasekaran, Sabine Weiskirchen, Ralf Weiskirchen

**Affiliations:** 1UT Southwestern Medical Center Dallas, Dallas, TX 75390-9014, USA; preethi.chandrasekaran@utsouthwestern.edu; 2Institute of Molecular Pathobiochemistry, Experimental Gene Therapy and Clinical Chemistry (IFMPEGKC), Rheinisch-Westfälische Technische Hochschule (RWTH) University Hospital Aachen, D-52074 Aachen, Germany; sweiskirchen@ukaachen.de

**Keywords:** lipocalins, cancer, structure, carrier proteins, transport proteins, LCN2, RBP4, A1M, PTGDS, human disease

## Abstract

The lipocalin proteins are a large family of small extracellular proteins that demonstrate significant heterogeneity in sequence similarity and have highly conserved crystal structures. They have a variety of functions, including acting as carrier proteins, transporting retinol, participating in olfaction, and synthesizing prostaglandins. Importantly, they also play a critical role in human diseases, including cancer. Additionally, they are involved in regulating cellular homeostasis and immune response and dispensing various compounds. This comprehensive review provides information on the lipocalin family, including their structure, functions, and implications in various diseases. It focuses on selective important human lipocalin proteins, such as lipocalin 2 (LCN2), retinol binding protein 4 (RBP4), prostaglandin D2 synthase (PTGDS), and α_1_-microglobulin (A1M).

## 1. Introduction

The term ‘lipocalin’ (LCN) was initially proposed by Syed Pervaiz and Keith Brew in 1987 to define a group of proteins that are significantly similar in amino acid sequence, disulfide bond arrangement, three-dimensional structure, functional similarities, and intron locations in their structural genes [1]. The initial members of these proteins included α1-glycoprotein (AGP), serum retinol binding protein (RBP or RBP4), β-lactoglobulin (BLG), α2_u_-globulin, and α_1_-microglobulin (A1M) [1]. These proteins were supposed to have a common role in the binding and transport of lipophilic molecules by their ability to fold into an eight-stranded β-barrel, which acts as an internal ligand binding site that can incorporate hydrophobic substances, such as retinol, and a single rod of an α-helix.

Since then, more than 1000 LCN genes have been identified in bacteria, fungi, plants, and animals, with 19 LCN genes found in the human genome [2]. Interestingly, among these 19 LCN genes, 16 cluster on human chromosome 9, while only RBP4, apolipoprotein D (ApoD), and apolipoprotein M (ApoM) are located on other chromosomes. This suggests that different lipocalins may have developed during evolution though gene duplication and divergence of an ancestral gene.

LCN1 and glycodelin (GD), a secretory immunosuppressive glycoprotein, are unique to humans. In contrast, LCN3, LCN4, LCN5, LCN11, LCN16, LCN17, and the 22 known functional major urinary protein (MUP) genes, classified as lipocalins, are exclusive to mice [2,3].

Lipocalins have evolved rapidly, resulting in 19 different proteins (Table 1). The sizes of these human proteins typically range from 160 to 200 amino acids in length with a molecular weight of about 18–40 kDa.

Despite their sequence diversity, the tertiary structure of lipocalins consists of a single, eight-stranded, hydrogen-bonded antiparallel β-barrel enclosing an internal cup-shaped ligand binding site. Their dynamic functions are attributed to their ability to bind a variety of small hydrophobic molecules, form complexes with macromolecules, and bind to specific cell surface receptors [4]. The ligand binding properties of lipocalins are highly specific and are associated with specific functions such as retinol binding protein-RBP transport in the retina, the removal of toxic molecules by LCN1, and the antioxidant functions of A1M. Additionally, the crucial role of lipocalins in binding xenobiotics should be considered when analyzing the pharmacokinetics and treatment outcomes of drugs [5,6,7].

Interestingly, lipocalins exhibit ligand-independent functions such as RBP4 activation of toll-like receptors 2/4 (TLR2/4) signaling in immune system cells, AM1 binding to heparin, and the antioxidant properties of bovine β-lactoglobulin [5,6,7,8]. Different lipocalins have varied glycosylation patterns that determine their interactions [8]. Lipocalins were originally characterized as transport proteins and storage proteins. Over the past years, it has become evident that lipocalins have diversified roles in the regulation of cell proliferation, differentiation, aging, cell death, inflammatory responses, reproduction, smell perception, metabolic disorders, cardiovascular remodeling, and cancer development and as carrier proteins in the clearance of intracellular and extracellular hydrophobic molecules [4,9].

In humans, lipocalins are found in plasma and body fluids, serving as carriers for a variety of small molecules. Additionally, LCNs are widely used as biochemical markers in human diseases. LCN1, also known as human tear pre-albumin, is one of the four major proteins in human tears [10,11]. It is secreted by lacrimal glands and acts as a lipid sponge on the ocular surface. Decreased levels of LCN1 have been linked to diabetic retinopathy and Sjogren’s disease. Another important human lipocalin, LCN2, plays a crucial role in mediating various inflammatory processes and is used as a biomarker for acute and chronic renal injury [12]. A unique group of lipocalin proteins, odorant-binding proteins 2A and 2B, are soluble carrier proteins that can bind odorants reversibly. OBP2A is highly expressed in nasal mucus, salivary glands, and lacrimal glands, while OBP2B is expressed in endocrine organs [13].

Another important human lipocalin, A1M, is used as a biomarker in proteinuria. Increased levels of A1M in urine indicate a defect in proximal tubules [14]. The lipocalins ApoD and ApoM have been shown to be altered in abnormal lipid metabolism [15]. A remarkable lipocalin in humans is C8G, the C8 gamma chain gene, one of the three subunits present in C8. C8 is a part of the membrane attack complex that participates in the irreversible association of complement proteins C6, C7, and C9, leading to the lyses of microbes [16].

Other important human lipocalins are orosomucoids (ORM1 and ORM2), which belong to the immunocalin subfamily. ORM1 and ORM2 were identified as predictive markers for systemic lupus, chronic inflammation, and rheumatoid arthritis [17]. A key human lipocalin is the progestogen-associated endometrial protein or glycodelin, a secreted immunosuppressive glycoprotein. Its downregulation leads to abortion in the first trimester due to increased stimulation of the immune system [18]. It is also expressed in several tumors, such as melanoma and lung cancers [19]. PTGDS (Prostaglandin D2 synthase) is involved in converting prostaglandin H2 to prostaglandin D2. Patients with attention deficit hyperactivity disorder are shown to have increased levels of PTGDS. An important human lipocalin is RBP4, which is the transporter of all-trans retinol. Increased RBP4 is found in insulin resistance, obesity, and non-alcoholic fatty liver disease [20].

As the lipocalin protein family is too large to be discussed exhaustively, this review focuses on three broad categories: homeostatic regulation of selected lipocalins in life processes such as aging, reproduction, and immune responses; a brief overview of the structure and characterization of the lipocalin family of proteins; and finally, the indispensable role of a few important lipocalins in a multitude of diseases. The scope of this review is limited to select important human lipocalins and their functions.

## 2. Structure and Sequence Properties of Lipocalins

The three-dimensional structure of the lipocalin family forms a calyx in a cylindrical manner, with a closed end on one side and an open end on the opposite side. The open end leads to a cavity for internal ligand binding and for transporting small hydrophobic compounds. Seven loops, L1–L7, connect the eight β-strands, coiled in a right-hand orientation around the central axis and interact through hydrogen bonds [21]. The first loop, L1, is large and flexible, serving as a dynamic omega-type lid for the open end of the barrel, while the other six loops are short hairpin-type-fashioned loops. Previous work has elucidated that the lipocalin fold is characterized by three structurally large, conserved regions: a short structurally conserved region (SCR)1-strand preceding a 3_10_-like helix, a SCR2-strand linking loop 6, and SCR3 connecting the last β-sheet with the C-terminal α-helix [22]. The loops L1, L3, L5, and L7 are located on one end of the β-barrel opening for the internal ligand binding site, while L2, L4, and L6 are at the other end, with the *N*-terminal polypeptide chain crossing this end of the barrel to close the end (Figure 1) [22].

The kernel lipocalins of the lipocalin (LCN) family share three conserved sequence motifs corresponding to three SCRs of the LCN fold. The outlier lipocalins of the LCN family share more than two SCRs and are more heterogeneous. This characterization classifies A1M, ApoD, complement C8, LCN2, L-PGDS, and RBP4 as kernel lipocalins, while AGP1, AGP2, LCN1, ApoM, LCN13, and LCN14 belong to the outlier category [2,21]. The diversified features of LCN protein binding receptors contribute to their functional heterogeneity. For example, LCN1 serves as a lipid sponge in the ocular surface, LCN2 binds TLR4 to mediate inflammatory responses, LCN7 regulates angiogenesis through TGF-β activation, and LCN13 and LCN14 regulate glucose homeostasis [8].

A signal peptide is present in all vertebrate lipocalins except for ApoM, which is cleaved during polypeptide synthesis in the rough endoplasmic reticulum. This is followed by their journey to the extracellular milieu through canonical secretion. All lipocalins are located extracellularly. However, many lipocalins, such as RBP and L-PGDS, internalize to cells and are degraded in lysosomes. They can also bind to mitochondria in damaged cells, such as A1M [5,23].

## 3. Lipocalins of Significance: An Overview

The various lipocalins exhibit significant diversity at the amino acid level. However, they all share a common tertiary structure consisting of an eight-stranded, antiparallel symmetrical β-barrel fold with a cylindrical shape. Additionally, most human lipocalins typically contain a C-terminal α-helix that is fully conserved (Figure 2).

### 3.1. α_1_-microglobulin

α_1_-microglobulin (A1M) is an antioxidant and tissue-cleaning protein that performs reductase, heme, and radical-binding functions. These properties are attributed to the strong electronegative surface-exposed thiol group of C34 on loop 1 at the open end of the lipocalin barrel. The A1M crystal structure has the characteristic lipocalin fold with a β-barrel and four loops at the open end, displaying a disulfide bridge between C72 and C169, as well as a free thiol on unpaired C34 [24]. It is important to note that C34 plays a role in one-electron oxidation and reduction reactions as well as in the binding and neutralization of target compounds [24].

Three molecular mechanisms contribute to the antioxidant properties of A1M. The capacity of A1M to reduce several biological substrates such as Met-Hb, cytochrome c, free iron, and the synthetic radical 2,2′-Azino-bis(3-ethylbenzothiazoline-6-sulfonic acid) is attributed to the free cysteine in position 34 and the three lysine residues in positions 92, 118, and 130. NADH, NADPH, and ascorbate serve as strong electron donors in this process. The second mechanism involves A1M’s ability to trap radicals in a reaction that utilizes the reducing potential of C34 [25,26]. A1M binds to heme and reacts with lysed red blood cells or heme-containing enzymes like myeloperoxidase [27].

A1M is a potential therapeutic agent proposed for use in pathological conditions involving free radicals and heme groups such as pre-eclampsia and hemolytic disorders due to its antioxidant properties [28,29]. Recently, A1M has been used as a biomarker for tubular damage as its levels are high in the kidneys and it possesses kidney-protective functions. Multiple animal studies have reported that A1M reduces damage induced by renal upregulation of stress genes and restores compromised renal function [30].

Another significant function of A1M is its erythroprotective effects, which hold promise in providing therapy for certain conditions. For instance, a preterm rabbit pup model of intraventricular hemorrhage showed decreased structural damage and reduced expression of pro-inflammatory genes with intracerebroventricular administration of A1M [29,31]. A1M also inhibits the destruction of collagen fibrils exposed to heme and repairs the damaged collagen fibrils. Interestingly, evidence of extracellular matrix repair mechanisms has been found in sheep, rabbits, and mice with induced pre-eclampsia-like symptoms. Treatment with A1M resulted in reduced damage to extracellular matrix structural components [32].

In summary, A1M exhibits reductase activity, radical scavenging activity, and heme-binding activity that as a whole help to protect cells and organs of oxidative stress-related damage (Figure 3).

### 3.2. Retinol Binding Protein 4

Retinol binding protein 4 (RBP4), another member of the lipocalin family, is a 21 kDa protein consisting of a single polypeptide chain with 201 amino acids. It serves as the major transporter of retinol, also known as vitamin A, in circulation. RBP4 is highly expressed in the liver, where retinyl esters are hydrolyzed to retinol and then bind to RBP4 in hepatocytes. When combined with transthyretin (TTR), the retinol/RBP4/TTR complex is released into circulation, allowing retinol to be delivered to tissues and bind to specific membrane receptors [5,33]. Sun et al. identified STRA6 as the cell surface receptor responsible for transporting retinol across the cell membrane. In addition, STRA6 facilitates the exchange between intracellular binding proteins and extracellular RBP4, helping to balance RBP4 import, export, metabolism, and storage [34].

RBP4 is associated with ocular diseases, impaired vision, dysregulation of lipid homeostasis, metabolic disorders, glucose intolerance, and cardiovascular disorders [35]. It has been reported that RBP4 mRNA expression increases in an mTOR-dependent manner in response to refeeding fasted mice [33]. A recent study found that loss of RBP4 in mice resulted in visual defects, even on a vitamin A-sufficient diet [36]. In the same context, transgenic expression of human RBP4 in muscle rescued serum retinol levels and suppressed visual defects caused by the loss of endogenous RBP4 [37]. Two missense mutations of RBP4, P.AT3T and P.A75T, were identified as causing autosomal dominant congenital eye malformations [38]. Additionally, rare mutations of RBP4 (C.248 + 1G > A) were reported in a patient with retinal dystrophy and ocular coloboma [39]. RBP4 mutations are also associated with osteoarthritis, acne vulgaris, and hypercholesterinemia [40]. Importantly, RBP4 polymorphisms are linked to coronary artery disease, cardiovascular disease risk factors, and obesity [41].

In 2005, Yang et al. reported an increase in RBP4 levels in the blood of insulin-resistant mouse models. This increase was observed in adipose tissue rather than the liver, leading to the hypothesis that RBP4 acts as an adipokine that links obesity and insulin resistance. More recently, it has been shown that within adipocytes, RBP4 triggers inflammatory responses by inducing the activation of macrophages, CD4 cells, and dendritic cells. This process involves NF-κB, JNK, and IL-1β signaling pathways independent of the association of RBP4 with retinol [42,43].

Interestingly, adipose tissue-specific overexpression of human RBP4 triggers hepatic steatosis and glucose intolerance despite the absence of alteration sin retinoid concentrations. Consistent with this, a high-fat diet aggravated RBP4-induced hepatic steatosis in these mice, indicating RBP4 stimulation of adipose tissue lipolysis and fatty acid release [44].

Another finding on RBP4 suggests that exposure to cold or low temperatures increases the plasma concentration of RBP4 and retinol in humans and mice. This is due to increased retinol mobilization from the liver into circulation for adaptations to low temperatures. In line with this, the loss of RBP4 in mice failed to induce thermogenic reprogramming of subcutaneous adipose tissue, making these animals more susceptible to cold temperatures [45].

Yet another fascinating finding is the decrease in systolic and diastolic blood pressure in RBP4 knockout mice, which protects these mice from angiotensin II-induced hypertension and cardiac hypertrophy [46].

A promising small molecule RBP4 ligand, A1120, has been identified. It disrupts the RBP4 and TTR complex by inducing conformational changes, leading to lower serum RBP4 levels. This ligand is also being investigated as a therapy for muscular dystrophy [40]. In summary, the role of RBP4 is versatile and can open new avenues for potential therapeutic targets for treating a wide spectrum of disorders.

Specifically, the impact of RBP4 on various pathways associated with physiological processes such as adipogenesis and gluconeogenesis, as well as disease-promoting activities like impairing insulin signaling and causing insulin resistance during the development of diabetes, make this lipocalin an intriguing therapeutic target for combating metabolic diseases (Figure 4).

### 3.3. Lipocalin 2

Lipocalin 2 (LCN2) is a 25 kDa circulatory protein consisting of 198 amino acids that plays a crucial role in transporting steroids, free fatty acids, prostaglandins, and hormones. It achieves this by binding to megalin/glycoprotein and GP330/SLC22A17 or 24p3R/LCN2R receptors. It is worth noting that the binding cavity of LCN2 is larger and more polar compared with other lipocalin proteins, allowing it to form large complexes and bind to large, less hydrophobic ligands to carry out essential functions in cell division and regulation. LCN2 is utilized as a biomarker for acute and chronic renal injury and is present in various cell types such as liver cells, renal cells, bone marrow, adipose tissue, lungs, and macrophages [48]. It is one of the 20 small secretory proteins involved in transporting hydrophobic ligands and has diverse biological functions, including anti-inflammatory and anti-bacterial properties. LCN2 also plays a role in cancer progression by breaking down the extracellular matrix and promoting metastasis. Furthermore, levels of LCN2 are elevated in cases of obesity and type 2 diabetes.

Multiple studies have reported that TNF-α, IFN-γ, and lipopolysaccharide (LPS) induce the expression of LCN2 in adipocytes and various other tissues. LCN2 expression is increased in the adipose tissue and liver of diabetic obese mice as well as in human diabetic individuals [49]. Few studies have reported increased LCN2 in obese human and murine models. The insulin-induced increase in LCN2 expression is suggested to be associated with the phosphoinositide-3-kinase (PI3K) and mitogen-activated protein kinase (MAPK) signaling pathways [50]. Molecules such as TNF-α, IL-1β, and NF-κB and their signaling pathways positively regulate LCN2 expression and are involved in the induction of insulin resistance [51].

LCN2 is also induced in the respiratory tract during inflammation, intestinal epithelium during inflammatory bowel disease, kidney during embryo development, and myocytes during bone formation [52]. The transient induction of LCN2 represents a physiological stress response, triggering tissue remodeling proliferation and repair.

The role of LCN2 in the regulation of iron homeostasis is well documented in multiple studies and is a potent therapeutic target in cancers [53]. Another predominant finding is the involvement of LCN2 in the periphery and brain in different central nervous system (CNS)-related conditions such as Alzheimer’s disease, Parkinson’s disease, and vascular dementia. LCN2 contributes to the pathophysiology of age-related central nervous system disorders by affecting cell death/survival signaling and iron metabolism and influencing inflammation [54]. LCN2 is known to act as a defense against bacterial infections by binding to iron-loaded bacterial siderophores and interfering with siderophore-mediated bacterial iron acquisition.

One study reported an increase in LCN2 in chronic kidney disease and left ventricular hypertrophy along with increased fibroblast growth factor 23 (FGF23). Although the knockout of LCN2 in chronic kidney disease mice did not show improvement in kidney function, serum FGF23 was reduced along with a marked improvement in survival [55].

Importantly, LCN2 is reported to be expressed in numerous pathological conditions in many organs and tissues, making this lipocalin a reliable diagnostic and prognostic biomarker for identifying disease severities or the progression of malignancies in different organs such as the liver, kidney, lungs, bone, brain, and heart (Figure 5). Consequently, the diagnostic, therapeutic, and clinical relevance of LCN2 are currently the focus of researchers and clinicians [56]. A recent systematic analysis identified 70 ongoing clinical trials utilizing LCN2 in the diagnostic and prognostic setting as a key outcome measure [57].

Another important feature of LCN2 is its role in regulating host responses to inflammation by maintaining iron homeostasis. Studies in *Lcn2* null mice have shown increased sensitivity to bacterial infections [58]. These bacteriostatic properties are mediated by the affinity of LCN2 for bacterial iron siderophores. Siderophores are a group of diverse, small, high-affinity iron-chelating compounds that are secreted by microorganisms in response to iron limitation in their environment to increase iron uptake [59].

Furthermore, LCN2 has been found to modulate hepatic lipid homeostasis by controlling the formation of intracellular lipid droplets. It accomplishes this by regulating members of the perilipin protein family, which are of fundamental importance for the formation of intracellular lipid droplets and the storage of triacylglycerol and cholesterol esters [60].

### 3.4. Lipocalin-Type PGD2 Synthase Protein

Lipocalin-type PGD2 synthase protein (L-PGDS), also known as prostaglandin D synthase (PTGDS), functions as a lipocalin protein that transports small lipophilic substances such as steroids, retinoids, thyroid hormones, bilirubin, and heme. Enzymatically, it aids in the production of prostaglandin D2 (PGD2) by catalyzing the isomerization of prostaglandin H2 (PGH2). The regulation of PGD2 receptors occurs through the D-prostanoid receptor (DP)1 and DP2 receptors. L-PGDS plays a crucial role in metabolic disorders such as diabetes mellitus, fatty liver disease, and obesity. L-PGDS is expressed in the central nervous system and human heart and is a major protein in human cerebrospinal fluid. PGD_2_, an important bioactive lipid mediator that regulates various pathophysiological functions, such as sleep, pain, food intake, immunity, cardiovascular disease, and reproduction, is produced by L-PGDS [61].

Lipocalin-type prostaglandin D synthase (L-PGDS) is a unique enzyme within the lipocalin family [62,63]. It is highly glycosylated lipocalin with two *N*-glycosylation sites and a molecular weight of 26,000. L-PGDS is a multifunctional lipocalin that binds to various lipophilic ligands such as PGD_2_, bilirubin, biliverdin, and retinoic acid [64]. Additionally, L-PGDS binds to amyloid β peptides, disaggregating amyloid β fibrils and acting as a major chaperone in human cerebrospinal fluid. The structure of L-PGDS consists of a typical lipocalin fold of a β-barrel with two sets of β-sheets, each consisting of four strands of anti-parallel β-sheets and a three-turn α-helix associated with the outer surface of the barrel. It has a hydrophobic central cavity that is larger than in other lipocalins. L-PGDS has two hydrophobic pockets, with one being the catalytic site known as site 1, which contains cysteine 65 (Cys65), corresponding to the ligand-binding pocket of other lipocalins. The other pocket is the non-catalytic site 2 [62,65].

It is important to understand the substrate-induced product release mechanism for L-PGDS. The apo form of L-PGDS has a wide opening through which PGH_2_ enters pocket 1 and binds to site 1. The cavity is then closed due to the interaction of PGH_2_ with the H2 helix, CD loop, and EF-loop of L-PGDS. This closed confirmation of L-PGDS holds the 9,11-endoperoxide group of PGH_2_ to interact with the thiol group of catalytic Cys65 [62]. The catalytic reaction produces PGD_2_, which still binds to site 1 with high affinity after catalytic activity. In the absence of PGH_2_, L-PGDS can act as a carrier of PGD_2_. If there is an excess of PGH_2_, the next molecule of PGH_2_ binds to site 2, inducing CD-loop movement to open site 1 and release PGD_2_. Then, PGH_2_ moves from site 2 to site 1, and this catalytic cycle starts again.

The Cys65 residue of L-PGDS is involved in its unique catalytic reaction not found in other lipocalins. It is interesting to note that the concentration of L-PGDS in serum, urine, and seminal fluid is used as a biomarker for various neurological disorders, sperm dysfunction, and renal disorders. Additionally, L-PGDS in the serum shows a circadian change, with a nocturnal increase and suppression during total sleep deprivation [62].

L-PGDS binds to many molecules with high affinities, such as all-trans and 9-cis-retinoic acid (K_D_ = 70–80 nM), thyroid hormones (K_D_ 0.7 to 2 µM), heme (K_D_ = 20–40 nM), and gangliosides GM1 and GM2 (K_D_ = 65–210 nM) [62]. It is known to scavenge harmful heme metabolites from cerebrospinal fluid and prevent the accumulation of harmful lipophilic substrates in lysosomal storage diseases and many other diseases [66,67,68].

PGD2 is extremely significant as an inflammatory mediator in allergies and is involved in both pro- and anti-inflammatory functions. For example, it has been reported that vascular hyperpermeability is accelerated in L-PGDS knockout mice with inflamed lungs, but this effect is suppressed by D-prostanoid receptor agonists. L-PGDS also exerts cardioprotective effects in several models. Interestingly, L-PGDS null mice show an increase in blood pressure and acceleration of thrombogenesis, indicating its role in controlling blood pressure and thrombosis.

Another important feature is that the deletion of L-PGDS promotes the degradation of cartilage in aging and increases the expression of extracellular degrading enzymes. This suggests a role of L-PGDS in protecting against age-related osteoarthritis. Strikingly, L-PGDS is expressed in endothelial cells of human melanoma and oral squamous cell carcinoma. L-PGDS gene deficiency accelerates angiogenesis, epithelial-to-mesenchymal transition, and tumor apoptosis. This suggests that PGD2 is a negative regulator of tumorigenic changes in tumor endothelial cells.

Furthermore, PGD2 has protective effects in safeguarding the endometrium against the development of adenomyosis. Double knockout mice of L-PGDS and hematopoietic prostaglandin D synthase (HPGDS) have been shown to develop adenomyotic lesions in their 6-month-old uterus. In summary, L-PGDS plays a dynamic role in multiple biological functions and is implicated in several diseases [66,67,68].

## 4. Implications of Lipocalins in Diseases

### 4.1. Lipocalins in Immune Responses

A predominant role of lipocalins is to act as an acute response protein upon infection, inflammation, and injury by synthesizing immune system mediators such as PGD2 and by controlling immunomodulators such as lysophosphatidylcholine (LPC). For example, LCN2 promotes inflammation, and A1M has immunosuppressive functions. Specifically, A1M inhibits key functions of IL-2 secretion. In monocytes, it inhibits oxidative burst and IL-1β production. AGP, LCN2, LCN1, and C8G (which generate a functional membrane attack complex) control bacterial infections by scavenging iron-containing siderophores. LCN1 is known to degrade microbial DNA [69,70,71,72]. This tear lipocalin (LCN1) along with secretory IgA, lysozyme, lactoferrin, and cystatin together make up most protein components of tear fluid. Additionally, tear lipocalin is demonstrated to have anti-microbial and anti-viral roles in the eye and mucous membranes of the upper airways. This is supported by LCN1 upregulation in the airways of patients with cystic fibrosis. The endothelial barriers to the passage of immune cells are regulated by AGP or ApoM. Interestingly, C8G and LCN2 play antagonistic roles in blood–brain barrier disruption. Noteworthily, lipocalins are involved in maintaining the composition and redox state of plasma membranes, lysosomal membranes, and many other cellular membranes [7,23,73,74].

Multiple studies have reported that α-1-glycoprotein (AGP) has numerous effects on all major leukocyte classes and platelets, and it exerts the direct stimulation of lymphocytes. Additionally, AGP stimulates superoxide anion generation from neutrophils and can stimulate or inhibit neutrophil aggregation. It is noteworthy that AGP induces the secretion of IL-1β, IL-6, and TNF-α. AGP is a heavily glycosylated lipocalin that undergoes carbohydrate modifications while exerting inflammatory effects [75].

### 4.2. Lipocalins in the Regulation of Metabolism

It is interesting to note that almost all lipocalins participate in the regulation of metabolic processes. The regulation of the insulin pathway by RBP4 or AGP, the involvement of A1M in obesity, endoplasmic reticulum stress, and the role of LCN2 and AGP in modulating food intake behavior are only some of the noteworthy features. Furthermore, RBP4, LCN2, and PGD participate in the metabolic response to cold stress. Indirect functions include the modulation of lipoprotein composition and functions by ApoD and ApoM [5,76].

LCN13, expressed in the liver, pancreas, epididymis, and skeletal muscle, is reported to regulate insulin sensitivity and nutrient metabolism. Circulating LCN13 levels are low in obese mice. Specifically, levels of circulating LCN13 are lower in the fasting state than in the fed state, suggesting the role of LCN13 in nutrient sensing and metabolic regulation. The predicted molecular mechanism of LCN13’s regulation of insulin is through binding to small hydrophobic molecules and controlling their transportation, stability, release, activation, and clearance. Interestingly, LCN13 may regulate insulin sensitivity by binding to and activating its cognate receptors on the plasma membrane. Similarly, LCN13 is predicted to bind to and activate its cognate receptors on hepatocytes, regulating liver lipid metabolism and fatty acid oxidation [77].

Similar effects have been reported with major urinary protein (MUP1) in modulating insulin sensitivity and glucose metabolism in obesity. The expression of MUP1 in the liver and plasma is significantly lower in mice on a high-fat diet that induces obesity. Additionally, MUP1 inhibits the expression of gluconeogenic genes and glucose production in hepatocytes [78].

In contrast, RBP4 promotes insulin resistance in skeletal muscle, adipocytes, and hepatocytes in addition to indirectly promoting insulin resistance by stimulating cytokines secretion by macrophages [79]. LCN2 also promotes insulin resistance like RBP4. The expression of LCN2 is increased in diabetic/obese mice in the adipose tissue and liver. LCN2 is reported to stimulate PPARγ, which mediates adipogenesis and lipogenesis in the liver and adipose tissue. In summary, LCN2 and RBP4 exert similar effects in insulin resistance and glucose metabolism, while LCN13 and MUP1 on the other hand exert opposite effects.

### 4.3. Lipocalins in Aging and Development

RBP4 is a major lipocalin involved in embryonic development, playing a role in the development of male and female organ–blood barriers through its receptor STRA6. Additionally, ApoD is involved in processes such as angiogenesis and chondrogenesis. Both ApoD and PGD are involved in myelin development. As organisms age, ApoD, ApoM, and PGD play a role in maintaining the function of specific organs such as the liver, brain, and cartilage [61,69,80].

Several studies have suggested that LCN2 contributes to various neuropathological processes in age-related central nervous system disorders, exacerbating neuroinflammation, cell death, and iron dysregulation, negatively impacting cognitive functions. Consistent with this, both systemic and central nervous system expression of LCN2 are increased in age-related central nervous system disorders, along with an increase in LCN2 with age [81].

Interestingly, elevated levels of mRNA and protein of LCN2 have been observed in the affected brain regions of individuals with Alzheimer’s disease. Additionally, increased levels of LCN2 have also been found in the substantia nigra of patients with Parkinson’s disease. Patients who have suffered from hemorrhagic stroke show an increase in serum LCN2 levels. Similarly, elevated LCN2 levels have been detected in human brain tissue and plasma following an ischemic stroke. Moreover, elevated quantities of LCN2 protein have been found in plaques within the brains of individuals with multiple sclerosis [82].

In summary, the effects of LCN2 on neuronal survival/death, neuroinflammation, brain iron metabolism, blood–brain barrier disruption, white matter damage, and neutrophil infiltration can play a versatile role in multiple central nervous system disorders such as Alzheimer’s disease, multiple sclerosis, and Parkinson’s disease.

### 4.4. Lipocalins in Reproduction

Lipocalins play a crucial role in modulating behaviors for reproductive success as well as in the development and function of reproductive organs. LCN8 and PGDS are essential for the development and maintenance of reproductive organs, while A1M prevents oxidative stress in the placenta and protects fertilization, implantation, and endometrial homeostasis. It is also worth noting that odorant-binding protein (OBP) controls microbiota and influences host physiology.

The crucial role of transporting retinoids for the complex process of spermatogenesis and differentiation of germ stem cells to motile spermatozoa is carried out by PTGDS, which is expressed in many tissues, including testes. An interesting study reported that LCN2, LCN6, and LCN8 were downregulated in caput sperm, while LCN2 was increased in corpus [83].

LCN5 is suggested to play a role in driving the development and maintenance of epididymal epithelium. In a study where LCN8 and FABP9 were knocked out in mice, abnormal sperm head and tail morphology were observed. Another significant role of LCN2 is in the transportation of reproductive hormone precursor cholesterol, indicating a connection to the endocrine system. Overall, the role of lipocalins is crucial in epididymal sperm maturation and processes related to capacitation and the acrosome reaction in the female reproductive tract [84].

### 4.5. Lipocalins in Human Cancer

The role of lipocalins in cell proliferation and differentiation has been extensively studied. In general, lipocalins mediate the regulation of cell differentiation by delivering their lipophilic ligands to specific cells or by acting as protease inhibitors against proteases involved in the progression of tumor cells. The immunosuppressive properties of lipocalins also contribute to tumor progression [85].

It is important to understand the role of specific lipocalins in human cancers. A1M is elevated in hepatocellular carcinoma but not significantly enough to be used as a clinical marker. In breast cancer cells, retinoic acid induces an increase in ApoD, and this elevated expression is a good prognostic marker in breast cancer. Interestingly, in prostate cancer cells, steroid activation of the androgen receptor reverses ApoD, leading to decreased proliferation. At high-steroid concentrations, ApoD is increased. In breast cancer, ApoD is reported to inhibit the translocation of phosphorylated MAPK into the nucleus, reducing the proliferation of cancer cells [85]. An interesting finding on oromucosoids is their presence in patients with leukemia. Altered glycosylation is a significant modification of oromucosoids in cancer in addition to an increase in the frequency of specific orosomucoid alleles in different cancer types.

LCN2 is an important lipocalin that is frequently associated with tumor size, stage, and invasiveness in multiple cancers. It plays a significant role in inhibiting apoptosis, stimulating proliferation, promoting epithelial-to-mesenchymal transition, and promoting angiogenesis [86,87,88]. LCN2 stabilizes the proteolytic enzyme MMP-9, preventing auto-degradation and promoting the metastasis of cancer cells. Furthermore, LCN2 expression is upregulated in pancreatic cancer, ovarian cancer, colorectal cancer, cholangiocarcinoma, and colon cancer. Additionally, cancer cell survival in the microenvironment is enhanced by its iron-shuttling functions. The interaction between LCN2 and catechol activates iron trafficking, regulates iron-responsive genes, and contributes to cancer progression. Cytokines secreted by the tumor microenvironment, such as IL-1β and TNF-α, stimulate the transcription of LCN2. Once expressed, LCN2 binds to catecholate to form a complex. This complex harvests iron from the extracellular matrix and enters the cell through LCN2 receptors. There are several mechanisms involved in the regulation of a variety of cancers by LCN2 [86,87,88]. In fact, many other lipocalins are involved in the modulation of different cancers. Due to the extensive information involved, we have only discussed the key features of a few lipocalins in important cancer types within the scope of our review.

In summary, several lipocalins are modulated in various types of human diseases, making them attractive targets for therapy. In addition to those discussed in our review, there are many other diseases associated with altered expression of lipocalins (Table 2).

Therefore, several lipocalins have attracted interest as biomarkers. However, it should be noted critically that further studies are necessary before introducing these new biomarkers in clinical practice [89].

## 5. Conclusions

Lipocalins are highly conserved small binding proteins expressed in various tissues, playing a versatile role in transporting several molecules. Additionally, lipocalins are extensively used as clinical biomarkers in various human diseases. Notably, lipocalins are involved in dynamic biological functions such as the modulation of immune response, prostaglandin synthesis, retinoid binding, mediating inflammatory reactions and metabolic processes and the regulation of cell homeostasis. However, the nature and extent of ligand bonding for a few of the human LCN family members need further investigation. It is clear that the understanding of the human LCN family has broadened in recent years, and further research will answer many intriguing questions about this interesting family of proteins. Although lipocalin knockout in animal models does not result in lethality, their role in optimizing various life processes is well understood. Therefore, it is important to investigate the role of lipocalins in physiological processes to improve lipocalin-based therapy for pathological conditions.

## Figures and Tables

**Figure 1 ijms-25-04290-f001:**
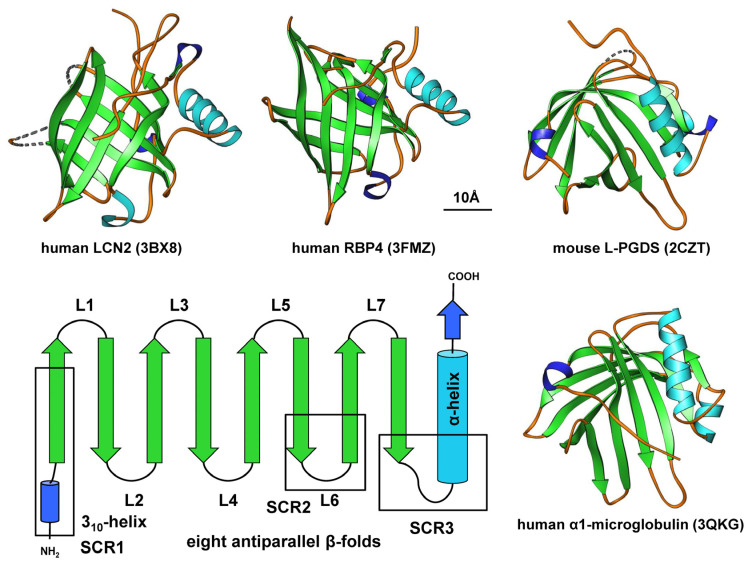
The lipocalin fold. Members of the lipocalin family are characterized by a typical β-barrel structure composed of eight-stranded, anti-parallel, symmetrical β-sheets (shown in green) linked by short loops (L1–L7). They also contain a stretch of α-helix (shown in light blue) and feature short structurally conserved regions (SCRs) and a short 3_10_-like helix at their *N*-terminal regions. The proteins depicted include human lipocalin 2 (LCN2), human retinol binding protein 4 (RBP4), mouse lipocalin-type prostaglandin D2 synthase (L-PGDS), and human α_1_-microglobulin (α1M). These structures were generated using Ribbons XP software (version 3.0) and the coordinates 3BX8, 3FMZ, 2CZT, and 3QKG deposited in the RCSB Protein Data Bank (http://www.rcsb.org, accessed on 19 March 2024). A size marker (10 Å) is provided.

**Figure 2 ijms-25-04290-f002:**
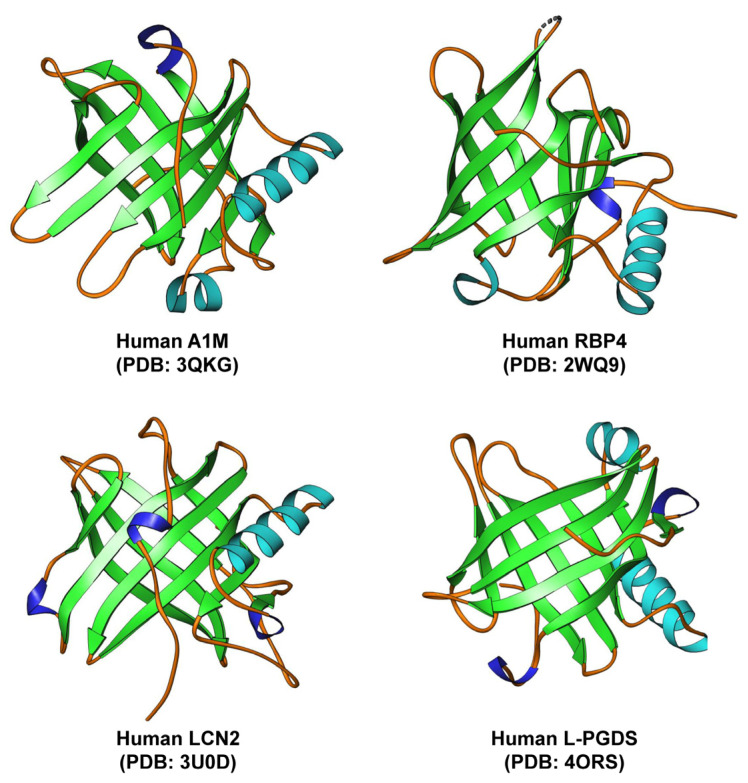
Structure of selected human lipocalins. Human α1-microglobulin (A1M), retinol binding protein 4 (RBP4), lipocalin 2 (LCN2), and lipocalin-type PGD2 synthase protein (L-PGDS) all share a highly similar structure with a typical eight-stranded, antiparallel symmetrical β-barrel fold. The structures of the discussed lipocalins were created using Ribbons XP software (version 3.0) and the coordinates 3QKG, 2WQ9, 3U0D, and 4ORS deposited in the RCSB Protein Data Bank (http://www.rcsb.org, accessed 6 April 2024).

**Figure 3 ijms-25-04290-f003:**
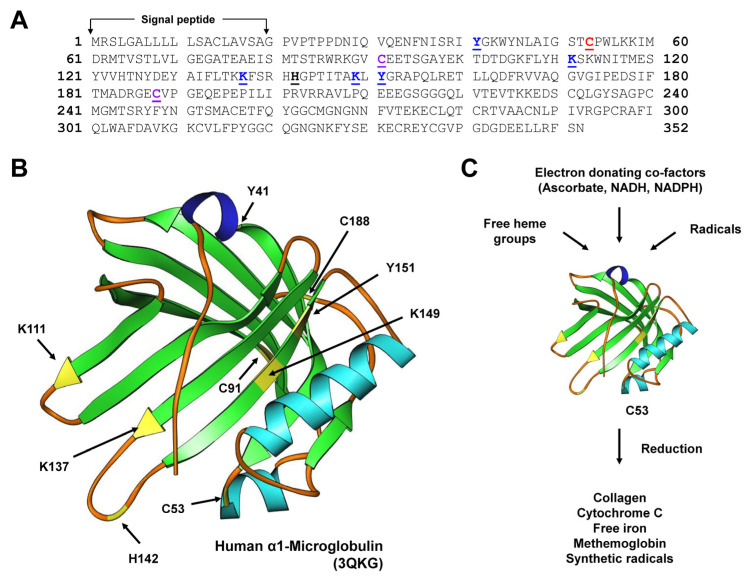
Physiological roles of α_1_-microglobulin (A1M). (**A**) The sequence of the human A1M preproprotein consists of 352 amino acids. The *N*-terminal part of the protein contains a 19-amino acid signal peptide that is cleaved during protein processing. The sequence was obtained from the NCBI Reference Sequence with access number NP_001624.1. Functional relevant amino acids are given in bold and are underlined. The relevance of these amino acids in explained in the legend to panel (**B**). (**B**) A1M possesses the typical lipocalin fold. The protein has the ability to protect cells and organs against oxidative stress-related injury. This activity is mediated by a strongly electronegative surface-exposed cysteine residue (C53 according to the position in the pre-pro-protein) located in loop 1 at the open end of the β-barrel. This residue can participate in one-electron oxidation and reduction conditions. Several targets for covalent modification (K111, K137, K149, Y41, and Y151), the heme binding residue (H142), and the residues forming the intramolecular disulfide bridge (C91, C188) are marked. In the structure, the mentioned functional relevant amino acids with selected functional activities are marked in yellow. (**C**) In particular, the free thiol group at position C53 allows A1M to reduce several biological targets (collagen, cytochrome C, free iron, methemoglobin) and synthetic radicals such as 2,2′-azino-bis(3-ethylbenzothiazoline-6-sulfonic acid). The reducing capacity of A1M depends on electron-donating co-factors such as ascorbate, nicotinamide adenine dinucleotide (NADH), nicotinamide adenine dinucleotide phosphate (NADPH), or other physiologically active electron-donating co-factors. For more details on the structure, functions, and physiological roles of A1M, refer to [6].

**Figure 4 ijms-25-04290-f004:**
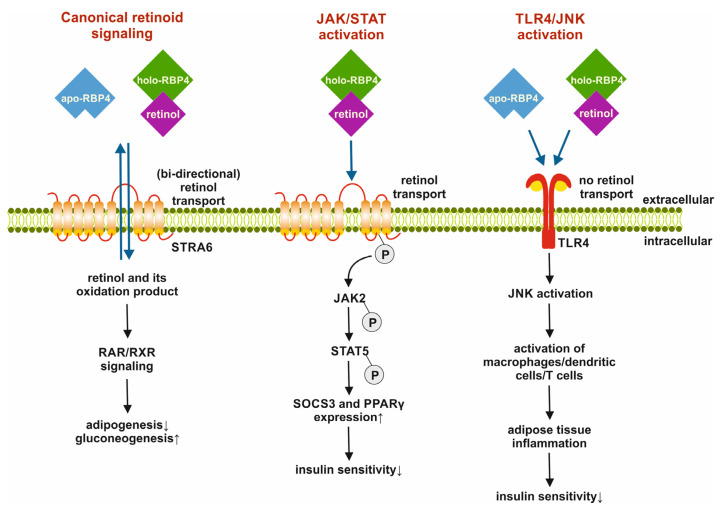
Metabolic activities of retinol binding protein 4 (RBP4). RBP4 can activate pathways related to canonical retinoid signaling and general cell metabolism. When RBP4 binds to the signaling receptor and transporter of retinol STRA6 (STRA6), it facilitates the transport of retinol across the cell membrane. This process mediates retinoic acid receptors/retinoid X receptors (RAR/RXR) signaling, which promotes gluconeogenesis and inhibits adipogenesis. The binding of holo-RBP4 can lead to the phosphorylation of STRA6, which then recruits and activates Janus kinase 2 (JAK2) and triggers signal transducer and activator of transcription 5 (STAT5) phosphorylation. This, in turn, induces proliferator-activated receptor γ (PPARγ) expression and lowers insulin sensitivity. Both apo- and holo-RBP4 can also trigger pro-inflammatory responses in immune cells by binding to toll-like receptor 4 (TLR4). This binding activates c-Jun N-terminal protein kinase (JNK), which is linked to the activation of macrophages, dendritic cells, and T cells. This activation can lead to inflammation in adipose tissue and reduced insulin sensitivity. In this figure, ↓ indicates inhibition, while ↑ marks activation of a process. The figure was adapted from a review [47] that delves deeper into RBP4 receptors and their functions in retinol transport and insulin signaling.

**Figure 5 ijms-25-04290-f005:**
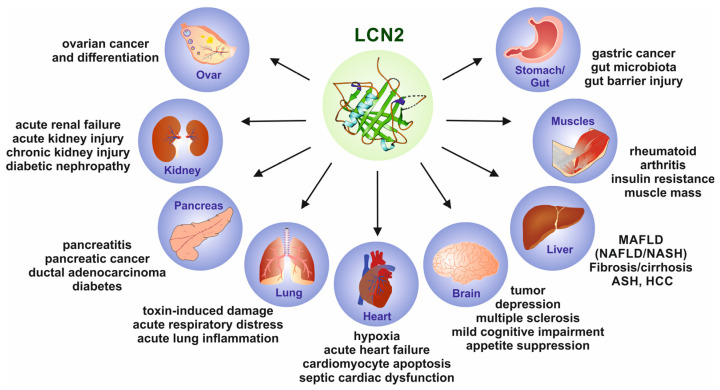
Lipocalin 2 (LCN2) is a biomarker that can identify inflammation, ischemia, infection, and acute organ damage. Changes in LCN2 levels may be caused by malignancies in various organs. The most significant associations are seen in inflammatory hepatic diseases, bowel disease, gastric cancer, thyroid cancer, pancreatic cancer, ovarian epithelial cancer, rheumatoid arthritis, and various kidney diseases. In addition, numerous diseases of the bladder, breast, skin, and other organs are known to be associated with altered LCN2 expression. For more information on experimental and clinical findings related to altered LCN2 expression, please refer to [56].

**Table 1 ijms-25-04290-t001:** Members of the human lipocalin family.

Protein	Alias	Chromosomal Localization	Predicted Function
LCN1	Tear lipocalin (TLC), tear prealbumin (TP), protein migration faster than albumin (PMFA); von Ebner gland protein (VEGP)	9q34	Removal of potentially harmful lipids, overproduced in infection ad stress, tear lipocalin
LCN2	Neutrophil gelatinase-associated lipocalin (NGAL), oncogenic lipocalin (24p3), uterocalin, MSFI, siderocalin (Scn)	9q34.11	Transport of small lipophilic substances, innate immunity, iron metabolism
LCN6	LCN5, hLcn5, UNQ643/PRO1273	9q34.3	Epididymal lipocalin involved in male fertility and single fertilization
LCN8	EP17, Epididymis secretory sperm binding protein, LCN5, Chromosome 9 open reading frame 137	9q34.3	Sperm maturation via transport of small hydrophobic molecules, male fertility, retinoid carrier protein within epididymis
LCN9	Epididymis luminal protein 129 (HEL129), epididdymal-specific lipocalin-9, 9230102I19Rik, MUP-like lipocalin	9q34.3	Binding small hydrophobic ligands
LCN10	9230112J07Rik, Epididymal-specific lipocalin-10	9q34.3	Protects against inflammation triggered vascular leakage.
LCN12	Epididymal-specific lipocalin-12, MGC48935, epididymis secretory sperm binding protein	9q34.3	Male fertility, binds all-trans retinoic acid within epididymis
LCN15	PRO6093, UNQ2541	9q34.3	Olfactory mucus, transport of vitamins, nucleosides
A1M	α_1_-microglobulin, α_1_-microglobulin/bikunin precursor (AMBP), heterogenous charge protein (HCP), complex-forming glycoprotein heterogeneous in charge, ITI, UTI, EDC1, HI30, ITIL, IATIL, ITILC, uristatin, uronic-acid-rich protein, trypstatin, growth-inhibiting protein 19, MGC64242, AI194774, DKFZp470D2211	9q32-q33	Antioxidant, heme binding, radical scavenging
PTGDS	Prostaglandin D2 synthase, PGD2, PDG2 synthase, (PDS, PGDS, PGDS2), lipocalin-type prostaglandin D synthase (LPGDS, L-PGDS)	9q34.2-q34.3	Catalyzes PGD_2_ production and transports lipophilic substances
ORM1	Orosomucoid 1 (ORM), α-1-AGP (AGP1), α- 1-glycoprotein α (AGP-A), HEL-S-153w	9q32	Tissue homeostasis and remodeling, acute phase reactant
ORM2	Orosomucoid 2, α-1-acid glycoprotein type 2 (AGP2), α1-glycoprotein β (AGP-B)	9q32	Acute phase reactant, immunomodulation and drug delivery, biomarker in cancers
OBP2A	Odorant-binding protein 2A, ddorant-binding protein (OBP), LCN13, OBP2C, OBPIIa, HOBPIIa	9q34	Scavenger of toxic odors, transport of hydrophobic molecules to olfactory receptors
OBP2B	Odorant-binding protein-2B, LCN14, OBPIIb	9q34	Chemosensory behavior
C8G	Complement component 8, γ subunit	9q34.3	Formation of membrane attack complex of the complement
PAEP	Progestagen-associated endometrial protein, glycodelin-S (GD-S), glycodelin-A (GdA); glycodelin-F (GdF), glycodelin-S (GdS), PEP; progestagen-dependent endometrial protein (PAEG), placental protein 14 (PP14)	9q34	Cell recognition, epithelial differentiation
RBP4	Retinol-binding protein-4, retinol-binding protein, retinal dystrophy iris coloboma and comedogenic acne syndrome protein (RDCCAS), microphthalmia/coloboma 10 (MCOPCB10)	10q23.33	Transport of the all-trans form of vitamin A
ApoD	Apolipoprotein D	3q29	Lipid metabolism, neuroprotection
ApoM	Apolipoprotein M, G3a, NG20, HSPC336	6p21	Anti-atherosclerotic, cholesterol efflux

**Table 2 ijms-25-04290-t002:** Lipocalins in human disease.

Lipocalin	Disease
LCN1	Decreased levels are associated with Sjogren’s syndrome, laser-assisted in situ keratomileusis (LASIK)-induced dry eye, and diabetic retinopathy. Increased expression is seen in cystic fibrosis.
LCN2	Increased expression in insulin resistance, obesity, and inflammatory processes.
A1M	Increased levels in proximal tubule defects. Upregulation of A1M protects the skin from damage caused by heme and reactive oxygen species.
ApoD	Increased expression associated with altered lipid metabolism, aging, and neurodegenerative diseases such as Parkinson’s disease and Alzheimer’s disease.
C8G	Deficiency of C8G is associated with rare recurrent infections of Neisseria meningitis. It controls bacterial infections by scavenging iron-containing siderophores.
PAEP	Decreased levels of PAEP are associated with first trimester abortion, while an increase is seen in gynecological malignancies, melanoma, and lung cancers.
PTGDS	Increase in attention deficit hyperactivity disorder and malignancies.
RBP4	Increased in obesity, insulin resistance, type 2 diabetes, and non-alcoholic fatty liver disease.
LCN13	Decreased expression in obesity and impact on liver lipid metabolism and fatty acid oxidation and insulin sensitivity.

Lipocalin

## Data Availability

This review only presents data that were previously published. No new data were generated.

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
