# Peer review of "Structure, Functions, and Implications of Selected Lipocalins in Human Disease"

_ijms, 2024, doi:10.3390/ijms25084290_

Round 1

Reviewer 1 Report

Comments and Suggestions for Authors

In the review “Lipocalins: Structure, Functions, and Implications in Disease” the authors consider four proteins of this family in detail, namely, lipocalin 2 (LCN2), Retinol binding protein 4 (RBP4), Prostaglandin D2 synthase (PTGDS), and α1-microglobulin (A1M). The authors also provide data on the involvement of lipocalins in various diseases.

In my opinion, the title is too general for the material in question. It should be stated in the title and abstract that only human lipocalins are considered. Although these proteins are also found in other living organisms. But the authors do not raise this issue in any way, focusing immediately on human lipocalins.

In the introduction, the place of human lipocalins among all representatives of this family should be determined. Reference 2 only considers human and mouse lipocalins. This is clearly not enough for a review.

Table 1 is not visually clear; the lines merge.

The colors in Figure 2B should be the same as in Figure 1. The indicated amino acid residues should also be highlighted in color or otherwise on the model.

Please define the abbreviations from the figure 3.

The section 3.4 is insufficient. It would be desirable to supplement this section with relevant information on the structure of the protein and corresponding molecules.

The authors consider separately four proteins related to lipocalins (the sections 3.1-3.4). One would like to see their similarities and differences. For example, figures comparing primary, tertiary structures.

The review will benefit from a general table of significant mutations in these proteins, as well as the diseases they cause and the ways they occur in the body.

The conclusions section does not contain any significant information. The section should be revised based on the text of the manuscript.

Authors are strongly encouraged to include an abbreviations section in their review.

Author Response

Dear Reviewer 1,

Thank you for taking the time to read our manuscript. We are grateful for your thoughtful comments. Please find attached a PDF-file in which we have summarized how we addressed your concerns and suggestions.

Regards

Ralf Weiskirchen

Reviewer 2 Report

Comments and Suggestions for Authors

I read with great interest this comprehensive review on the Role of Lipocalins and their implication on various diseases. I agree with the structure of the paper proposed by the authors.

However, I suggest the following changes in order to improve the overall quality of the paper.

1) a paragraph on the role of lipocaline and cancer is missing. Authors should address more deeply the role of lipocaline and tumor progression. 

2) Often authors failed to add references to their statements. please revise this point.

e.g. on page 8 line 238 add more references.I suggest to rely on: 10.23736/S2724-6051.21.04308-1

3) minor typos and grammar

Author Response

Dear Reviewer 2,

Thank you for taking the time to read our manuscript. We are grateful for your thoughtful comments. Please find attached a PDF-file in which we have summarized how we addressed your concerns and suggestions.

Regards

Ralf Weiskirchen

Round 2

Reviewer 1 Report

Comments and Suggestions for Authors

Most of the items have been resolved. I believe that the article can be published.

I recommend expanding the abbreviations section by adding other abbreviations from the article.

Reviewer 2 Report

Comments and Suggestions for Authors

authors addressed my major concerns. I endorse publication